# Acute kidney injury – A frequent and serious complication after primary percutaneous coronary intervention in patients with ST-segment elevation myocardial infarction

Abdellatif El-Ahmadi[1][ORCID]*, Mujahed Sebastian Abassi[1][ORCID], Hedvig Bille Andersson[1], Thomas Engstrøm[1], Peter Clemmensen[2,3], Steffen Helqvist[1], Erik Jørgensen[1], Henning Kelbæk[4], Frants Pedersen[1], Kari Saunamäki[1], Jacob Lønborg[1], Lene Holmvang[1]

1 Department of Cardiology, Rigshospitalet, Copenhagen University Hospital, Denmark, 2 Department of General and Intervention Cardiology, University Heart Center, Hamburg-Eppendorf, Germany, 3 Department of Medicine, Division of Cardiology, Nykoebing-Falster Hospital, University of Southern Denmark, Odense, Denmark, 4 Department of Cardiology, Zealand University Hospital, Denmark

☯ These authors contributed equally to this work.
* a.elahmadi@gmail.com

## Abstract

### Objectives

The aim of the study was to investigate the incidence, risk factors and long-term prognosis of acute kidney injury (AKI) in patients with ST-segment elevation myocardial infarction (STEMI) treated with primary percutaneous coronary intervention (primary PCI).

### Method

A large-scale, retrospective cohort study based on procedure-related variables, biochemical and mortality data collected between 2009 and 2014 at Rigshospitalet, Copenhagen, Denmark. AKI was defined as an increase in serum creatinine of 25% during the first 72 hours after the index procedure.

### Results

A total of 4239 patients were treated with primary PCI of whom 4002 had available creatinine measurements allowing for assessment of AKI and inclusion in this study. The mean creatinine value upon presentation for all patients was 84 μmol/l (standard deviation (SD) ±40) and 97 μmol/l (SD ±53) at peak. AKI occurred in a total of 765 (19.1%) patients. Independent risk factors for the occurrence of AKI were age, time from symptom onset to procedure, peak value of troponin-T, female sex and the contrast volume to eGFR ratio. In a multivariable adjusted analysis AKI was independently associated with a higher mortality rate at 5 years follow-up (hazard ratio 1.39 [95%-confidence interval 1.03–1.88]).

**Data Availability Statement:** All data files are available from the Figshare database from this link: https://doi.org/10.6084/m9.figshare.10277648.v1.

**Funding:** The author(s) received no specific funding for this work.

**Competing interests:** The authors have declared that no competing interests exist.

## Conclusion

In STEMI patients treated with primary PCI one in five experiences acute kidney injury, which was associated with a substantial increase in both short- and long-term mortality.

## Introduction

Timely treatment with primary percutaneous coronary intervention (primary PCI) is the recommended treatment for patients with ST-segment elevation myocardial infarction (STEMI) [1]. The prognosis for patients with STEMI has improved during the last decades, and the 1-year mortality is now around 11% [2]. However, some patients may still have an unfavorable outcome. Acute Kidney Injury (AKI) following PCI is a common complication that is observed in >15% of STEMI patients treated with primary PCI [3,4]. The incidence of AKI is higher following a primary PCI than after elective PCI suggestively owing to the use of more contrast agent during the procedure [5,6]. AKI may occur in patients with both preexisting renal disease and in patients with normal baseline renal function [7]. In STEMI patients the development of AKI following primary PCI has previously been related to an increased in-hospital stay and 3-year mortality [8,9]. Most previous studies on the incidence, risk factors and prognostic importance of AKI in patients with STEMI treated with primary PCI have included patients from randomized studies with various in- and exclusion criteria. Thus, these studies do not represent the full spectrum of patients with STEMI and there are no data on the importance of AKI in an all-comer STEMI population and on the mortality beyond 3-years. Therefore, the aims of the present study were to:

1. Report the incidence of AKI among an all-comer STEMI population undergoing primary PCI

2. Identify the risk factors for the development of AKI after primary PCI in STEMI patients

3. Report the long-term effects of AKI on mortality in a non-selected all-comer STEMI population.

## Methods

### Study population

From the Eastern Danish Heart Registry, a clinical registry with prospective registration of all PCI procedures performed in Eastern Denmark, consecutive patients undergoing primary PCI due to STEMI from November 1, 2009 to December 31, 2014 at a large tertiary invasive center were identified. A diagnosis of suspected STEMI was made by the cardiologist on call and based upon symptoms and electrocardiogram (ECG) as specified in the current guidelines [1]. Patients were included in the present study only of symptoms onset to PCI was<12 hours from symptom onset. Only patients treated with primary PCI after the coronary angiography were included in the present study. The study was approved by the Danish Data Protection Agency (case file no 30–1441) and the Danish National Board of Health (case file no 3-3013-1210/1).

### Definition of AKI

AKI was defined according to the European Society of Urogenital Radiology guidelines as an increase of 25% in serum creatinine from baseline creatinine within 72 hours from contrast media administration [4].

## Data collection

Creatinine values, procedure related variables and variables related to other risk factors related to kidney or coronary artery disease were obtained from hospital charts. Baseline blood samples were obtained at admission before primary PCI. Follow-up blood samples for creatinine were obtained between 24 and 72 hours after admission. If more than one creatinine value were present, the highest creatinine value was used for AKI analysis. Vital status was obtained from the Danish Centralized Civil Registry and the Cause of Death Registry using the CPR-number. Mortality has a very low risk of misclassification in the Danish registries and was used for this reason. Patients without a CPR-number (i.e. tourists) were excluded from this analysis due to missing follow-up data. Patients were followed from the first day of their primary admission until they either died or the follow-up period ended on the 15th of May 2015.

The assay for troponin changed in the study period, making high sensitivity measurement available, lowering the reference interval from 50ng/L to 14 ng/L.

## Statistical analysis

Patients who developed AKI after the procedure were compared to patients without AKI. The comparison was done using χ2 –test or Fisher´s exact test for categorical variables and Student T-test or the Mann-Whitney for continuous variables. To identify independent predictors for the development of AKI variables with $p < 0.05$ by univariate analysis were included in a multivariate logistic regression model. Finally, backward elimination was done by removing the variable with the highest p-value for each step. The mortality rates were assessed visually by a Kaplan-Meier plot and statistically using a log-rank test. The hazard ratio for mortality was calculated by Cox-regression. In order to identify predictors for mortality a multivariate Cox regression analysis was performed using all variables with significant association ($p < 0.05$) to mortality in the univariate analysis. Backward elimination was done by removing the variable with the highest p-value for each step. The assumption of the Cox proportional hazards regression models was tested by plot of cumulative sum of martingale-based residuals and found valid. Both regression analyses were performed using backward elimination. No imputations were done regarding missing values. A two-sided p-value $< 0.05$ was considered statistically significant and the data were analyzed using the statistical software IBM SPSS Statistics version 22.0 (SPSS Inc., Chicago, IL, USA).

## Results

### Baseline characteristics

A total of 4239 patients were treated with primary PCI between November 1, 2009 and December 31, 2014 of whom 4002 (94%) had available creatinine measurements allowing for assessment of AKI and thus inclusion in this study. A total of 765 patients (19.1%) developed AKI within 72 hours after primary PCI. Differences in the baseline characteristics between patients with and without AKI are summarized in Table 1. In brief, patients with AKI were older, more often female, and with lower left ventricular ejection fraction (LVEF), lower estimated glomerular filtration rate at admission (eGFR) as well as higher contrast volume/eGFR ratio. Patient with AKI also had a higher frequency of cardiogenic shock and pre-admission cardiac arrest. Both time from symptom onset to primary PCI and duration of the procedure were longer for patients that developed AKI.

### Independent predictors of AKI

In the multivariate backward elimination logistic model, the independent predictors for the development of AKI were: Peak TnT, symptom-to-PCI time, female sex and contrast volume

**Table 1. Baseline characteristics of AKI vs. non-AKI.**

| Risk factor | AKI | No-AKI | P value |
|---|---|---|---|
| Age, (SD) | 66 (13) | 63 (12) | < 0.001 |
| Female sex, (%) | 222 (29%) | 806 (25%) | 0.019 |
| BMI, kg/m$^2$, (SD) | 27 (7.3) | 27 (8.6) | 0.018 |
| Diabetes, (%) | 88 (12%) | 385 (13%) | 0.88 |
| Hypertension, (%) | 289 (41%) | 1171 (39%) | 0.26 |
| Smoker, (%) | 457 (72%) | 2215 (76%) | 0.016 |
| Extra cardiac arteriopathy, (%) | 30 (4.0%) | 124 (3.9%) | 0.85 |
| Family history of ischemic heart disease, (%) | 195 (32%) | 976 (36%) | 0.034 |
| Hyperlipidemia, (%) | 182 (28%) | 813 (29%) | 0.67 |
| History of MI, (%) | 72 (10%) | 276 (8.9%) | 0.32 |
| Previous PCI, (%) | 58 (7.6%) | 303 (9.4%) | 0.14 |
| Previous CABG, (%) | 20 (2.6%) | 50 (1.6%) | 0.039 |
| History of stroke, (%) | 47 (6.1%) | 139 (4.3%) | 0.022 |
| History of heart failure, (%) | 40 (5.5%) | 81 (2.6%) | < 0.001 |
| Cardiogenic shock pre-procedure, (%) | 38 (5.4%) | 74 (2.4%) | < 0.001 |
| Cardiac Arrest pre-procedure, (%) | 78 (11%) | 204 (6.7%) | < 0.001 |
| Contrast volume, mililiters, (SD) | 114 (62) | 111 (60) | 0.15 |
| Procedure length, minutes, (SD) | 38 (24) | 34 (20) | < 0.001 |
| Creatinine on admission, μmol/l, (SD) | 81 (43) | 84 (39) | 0.08 |
| Creatinine peak, μmol/l, (SD) | 125 (83) | 91 (41) | <0.001 |
| eGFR at admission, ml/min, (SD) | 67 (28) | 76 (18) | < 0.001 |
| Contrast volume/eGFR ratio, (SD) | 2.5 (3.0) | 1.6 (1.3) | < 0.001 |
| LVEF, %*, (IQR) | 40 (30–50) | 45 (35–50) | < 0.001 |
| Peak Troponin T, mg/l*, (IQR) | 5.7 (2.2–12) | 3.0 (1.1–6,3) | < 0.001 |
| Time to pPCI, hours*, (IQR) | 3.4 (2.3–5.4) | 2.8 (2.1–4.4) | < 0.001 |

AKI = Acute Kidney Injury; BMI = Body Mass Index; MI = Myocardial Infarction; pPCI = Primary Percutaneous Coronary Intervention

CABG = Coronary Artery Bypass Graft; LVEF = Left Ventricular Ejection Fraction, eGFR = Estimated Glomerular Filtration Rate

SD = ± Standard Deviation, IQR = Interquartile Range

*Median

(%) is percentage of total for each group

Creatinine peak is within 3 days after time of admission

Time to PCI is from symptom onset to balloon dilatation and is measured in hours

Cardiac arrest pre-procedure refers to patients who have been successfully resuscitated

LVEF: Values were only present for 993 out of 4002 patients.

to eGFR ratio, whereas age and smoking were statistically insignificant in the last step of the analysis. The analysis of independent predictors is summarized in Table 2. Since, LVEF at admission was only available in 24.8% of the patients it was excluded from the multivariable analyses.6

**Table 2. Multivariate analysis of the significant risk factors for AKI.**

| Multivariate | | | | Backward elimination | | |
|---|---|---|---|---|---|---|
| Risk factors for AKI | HR | 95% CI | p value | HR | 95% CI | p value |
| Age, per year | 1.01 | 0.99–1.02 | 0.08 | 1.01 | 1.00–1.02 | 0.07 |
| Female sex | 1.28 | 1.01.-1.63 | 0.041 | 1.28 | 1.02–1.62 | 0.043 |
| BMI, per unit | 0.99 | 0.97–1.01 | 0.31 | - | - | - |
| Smoker | 0.80 | 0.63–1.02 | 0.06 | 0.82 | 0.65–1.03 | 0.09 |
| Family history of ischemic heart disease | 0.98 | 0.79–1.23 | 0.91 | - | - | - |
| Previous CABG | 1.14 | 0.46–2.82 | 0.78 | - | - | - |
| History of stroke | 1.24 | 0.76–2.04 | 0.39 | - | - | - |
| History of heart failure | 1.47 | 0.82–2.61 | 0.19 | - | - | - |
| Cardiogenic shock pre-procedure | 1.45 | 0.61–3.47 | 0.41 | - | - | - |
| Cardiac Arrest pre-procedure | 0.83 | 0.46–1.48 | 0.52 | - | - | - |
| Procedure length, per minute | 1.00 | 0.99–1.00 | 0.23 | - | - | - |
| eGFR at admission, per ml/min | 1.01 | 1.00–1.01 | 0.18 | - | - | - |
| Contrast volume/eGFR ratio, (SD) | 1.14 | 1.04–1.25 | 0.006 | 1.08 | 1.02–1.16 | 0.015 |
| Peak Troponin T, mg/l | 1.08 | 1.06–1.10 | <0.001 | 1.08 | 1.06–1.10 | <0.001 |
| Time to PCI[1], per hours | 1.09 | 1.06–1.13 | <0.001 | 1.09 | 1.05–1.13 | <0.001 |

AKI = Acute Kidney Injury; BMI = Body Mass Index

CABG = Coronary Artery Bypass Graft; IHD = Ischemic Heart Disease; PCI = Percutaneus Coronary Intervention

eGFR = Estimated Glomerular Filtration Rate

[1]Time to PCI is from symptom onset to balloon dilatation and is measured in hours.

LVEF: Due to missing values this variable have been left out in the analysis.

## Mortality and AKI

Throughout the follow-up period of 3.4 years (range 0–5.52 years) a total of 527 (13.2%) patients died. The presence of AKI was statistically significant associated with higher mortality rates compared to patients without AKI during the entire follow-up period and at each landmark analysis, Table 3 and Fig 1. A landmark analysis did also show that the occurrence of AKI was related to mortality from 30 days to end of follow-up (HR 1.70 (95% 1.31–2.23)). Moreover, AKI remained independently associated with long-term mortality in multivariate Cox regression models adjusting for variables found to be associated with mortality, Table 4. Other important factors associated with long-term mortality were age, previous CABG, history of stroke, history of heart failure, cardiogenic shock at admission, resuscitated cardiac arrest, peak TnT and symptom-to-PCI.

## Discussion

In an unselected all-comer population AKI occurred in one in five patients within the first 72 hours after primary PCI and was strongly related to both short and long-term prognosis. In

**Table 3. Mortality.**

| Period | AKI | No-AKI |
|---|---|---|
| 30 days, (%) | 103 (13.5%) | 146 (4.5%) |
| 1 year, (%) | 142 (18.6%) | 234 (7.2%) |
| End of follow up, (%) | 178 (23.3%) | 349 (10.8%) |

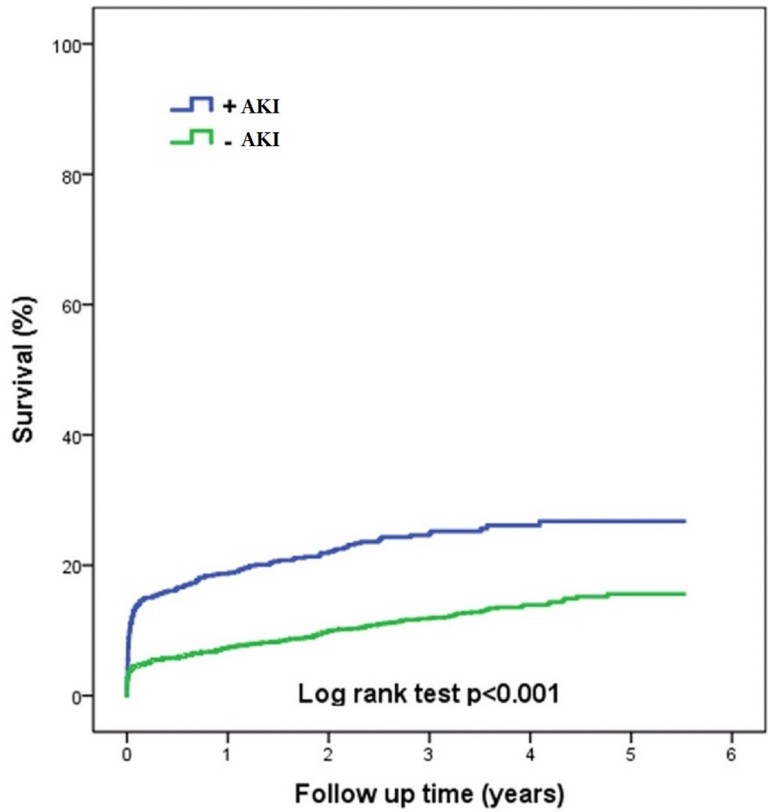

**Fig 1. Survival.**

previous studies the incidence of AKI following primary PCI in STEMI varied between 15–30% depending on study population, design and definition of AKI [3,4,8]. In comparison the incidence of AKI in the general population after contrast media consuming procedures (not restricted to PCI) is estimated to be only 1–2% [10]. There is no global consensus on the definition of AKI and in the present study we used the definition recommended in the ESUR guidelines which is an increase of 25% in serum creatinine from baseline creatinine within 72 hours from contrast media administration [4].

In the "Kidney Disease Improving Global Outcomes" criteria (KDIGO) there are 3 stages of AKI, stage 1 using 1,5–1,9 increase from baseline creatinine and stage 3 using more 3 times increase from baseline creatinine, and we translate this as a severity score, therefore 25% increase could presumably be the first point in developing the AKI and gradually getting worse with increasing creatinine levels [11].

The higher mortality rates for AKI patients is a consistent finding across studies [12]. A sub-study from the HORIZONS-AMI study found a mortality rate of 8.0% at 30 days and 16.2% after 3 years of follow up, as compared to 0.9% and 4.5% respectively for patients with and without AKI [8]. In a similar study a sevenfold increase in mortality (23.3% vs. 3.2%) was seen at 1 year [9]. In the present study AKI was related to a 39% relative increase in all-cause mortality through a median observation time of 3.4 years.

Thus, our results confirm previous findings, but add important data regarding long-term mortality and demonstrate the clinical relevance of AKI in an all-comer STEMI population including patients with out-of-hospital cardiac arrest and cardiogenic shock [13]. The reason for poorer outcome following AKI is still unknown. It may be due to preexisting illness, mainly

**Table 4. Multivariate analysis of the significant risk factors affecting mortality.**

| Multivariate | | | | Backward elimination | | |
|---|---|---|---|---|---|---|
| Risk factor | HR | 95% CI | p value | HR | 95% CI | P value |
| AKI | 1.38 | 1.02–1.87 | 0.037 | 1.39 | 1.03–1.88 | 0.033 |
| Age, per year | 1.06 | 1.04–1.07 | <0.001 | 1.06 | 1.05–1.07 | <0.001 |
| Female sex | 1.02 | 0.75–1.37 | 0.92 | - | - | - |
| BMI, per unit | 0.99 | 0.97–1.02 | 0.63 | - | - | - |
| Smoker | 1.33 | 0.98–1.81 | 0.07 | 1.31 | 0.97–1.77 | 0.076 |
| Family history of ischemic heart disease | 0.85 | 0.63–1.16 | 0.32 | - | - | - |
| Previous CABG | 2.31 | 1.15–4.66 | 0.019 | 2.16 | 1.09–4.31 | 0.028 |
| History of stroke | 1.65 | 1.06–2.57 | 0.026 | 1.61 | 1.04–2.50 | 0.033 |
| History of heart failure | 2.49 | 1.52–4.07 | <0.001 | 2.44 | 1.51–3.96 | <0.001 |
| Cardiogenic shock pre-procedure | 2.87 | 1.53–5.40 | 0.001 | 2.66 | 1.45–4.90 | 0.002 |
| Cardiac Arrest pre-procedure | 3.13 | 1.93–5.09 | <0.001 | 3.04 | 1.88–4.92 | <0.001 |
| Procedure length per minute | 0.99 | 0.98–1.00 | 0.20 | - | - | - |
| eGFR on admission per ml/min | 0.99 | 0.98–0.99 | <0.001 | 0.99 | 0.98–0.99 | <0.001 |
| Contrast volume/eGFR ratio | 1.04 | 0.97–1.12 | 0.27 | - | - | - |
| Peak Troponin T, per 1mg/ml | 1.03 | 1.01–1.04 | 0.001 | 1.03 | 1.01–1.04 | 0.001 |
| Time to PCI[1], per hour | 1.06 | 1.02–1.10 | 0.003 | 1.06 | 1.02–1.10 | 0.003 |

AKI = Acute Kidney Injury; BMI = Body Mass Index

CABG = Coronary Artery Bypass Graft, MI = Myocardial Infarction.

PCI = Percutaneous Coronary Intervention.

[1]Time to PCI is from symptom onset to balloon dilation and is measured in hours.

LVEF: Due to missing values this variable have been left out in the multivariate analysis.

of either renal or atherosclerotic character [14]. Another contributing factor may be damage to other organs during acute renal failure. Previous studies found severely diminished coronary vascular tone, reserve and reactivity after acute renal failure [11].

The present study is a true all-comer population including patients with an inherent risk of nephropathy such as patients with cardiogenic shock and cardiac arrest. These patients are almost always excluded from randomized clinical trials. In the recently published CULPRIT--SHOCK trial, the need for renal replacement therapy were as high as 11.6–16.4% among the study participants, underlining that nephropathy is a common complication in patients with acute myocardial infarction and shock [15]. Cardiogenic shock at admission and cardiac arrest before the procedure were not independently associated with AKI in the present study, possibly explained by the low frequency of both cardiac arrest and cardiogenic shock, and the fact that the most critically ill patients die before additional blood work-up can be performed.

Previously other risk factors for AKI have been reported such as old age, diabetic nephropathy, time from onset of symptoms to primary PCI, preexisting renal disease, congestive heart failure, contrast volume and length of the procedure [10,16][6,8,17],[18]. And in a recent study with 1656 STEMI patients both congestive heart failure and reduced left ventricular ejection fraction were identified as individual predictors of AKI [19].

In the present study age, female sex, time to treatment, peak TnT level and the contrast volume to eGFR ratio were all independent predictors for the development of AKI. Contrast volume, preexisting diabetes and procedure length were not independently associated with AKI

in the present population may be explained by several factors. Earlier studies reported much higher average use of contrast media volume among patients developing AKI(153–378 ml) compared to the 111–114 ml used in the present study [6,8,17]. Although some studies have identified diabetes as an independent risk factor other studies have not [6,7,17,20]. One study indicates diabetes only being a risk factor combined with renal insufficiency but not per se [21]. Similar circumstances could explain the reason that the contrast to eGFR ratio is an independent predictor of the development of AKI while contrast volume is not.

That diabetes mellitus was not associated with AKI in the present study may also be due to the low prevalence of diabetes in the study population in combination with the relatively small amount of contrast used. A recent study found admission hyperglycemia to be an independent risk factor for AKI following primary PCI [22].

The association between time to treatment and AKI could be explained by the changes in renal hemodynamics that occurs during an acute myocardial infarction [18]. Also, longer ischemia times may lead to larger infarcts and subsequent heart failure, which also explains why infarct size determined by peak troponin T was associated with AKI.

Controversies still remain over the effectiveness of interventions to prevent AKI, e.g. hydrating with intravenous fluid (IV) and administering drugs like *N*-Acetylcysteine or Fenoldopam Mesylate [23,24,25,26]. Although most studies state no benefit from these interventions, a few randomized trials have found hydration with IV normal saline to be consistently effective. Mueller et al performed a large prospective randomized study comparing isotonic IV saline (0.9%) with half-isotonic IV saline (0.45%) and concluded that isotonic saline (0.9%) was superior in terms of reducing AKI. Unstable cardiac patients undergoing primary interventions were the most likely to benefit from isotonic hydration [27]. Starting the hydration earlier and continuing longer are considered best practice. However, no single regimen of IV saline has yet been established in practice [28]. A recent single-center retrospective study suggested that a micro axial percutaneous left ventricular assist device may protect against acquired kidney injury during high risk PCI. The majority of the patients in that study had acute coronary syndromes but only 13% were true STEMI[29]. Finally, lowering the patient's blood glucose levels before procedure have been suggested by some authors [22].

A recent review summarizes the pathophysiology and highlights preventative strategies for patients undergoing angiographic procedures. It also confirms that diabetes mellitus has not been shown to be an independent risk factor. It states that contrast volume > 350ml or repeated administration within 72 hours was shown to be associated with an increased risk. The review states that developing AKI is correlated with accelerated progression of an underlying chronic kidney disease. The review concludes that additional studies is needed to effectively determine the true toxic effect and to evaluate the survival benefit of preventing AKI [30].

## Study limitations

This is a single-center observational study and although there is a quality in its all-comers design, treatments and patients may differ from other primary PCI centers. AKI could only be diagnosed if at least two blood samples for creatinine measurements were available. Thus, patients dying very early after admission were not included in the study. Only data on all-cause mortality were available.

Moreover, given the observational nature of this study no certainty in terms of causality between AKI and adverse outcome can be made. Creatinine values beyond 72 hours were not available and thus we do not have any information on the long-term renal function and its impact on mortality. Information regarding left ventricular function at admission was not

available in a large proportion of the patients due to the emergency situation. Finally, retro-spective studies are susceptible to the issue of missing data, so too in this study, which intro-duces the risk of selection bias.

Since the assay for TnT changed during the study period, there is a possibility that very low but elevated values could have be overseen.

## Conclusion

In STEMI patients treated with primary PCI one in five experiences acute kidney injury, which was related to substantial increase in both short- and long-term mortality.

## Author Contributions

**Conceptualization:** Lene Holmvang.

**Formal analysis:** Jacob Lønborg, Lene Holmvang.

**Investigation:** Jacob Lønborg.

**Methodology:** Jacob Lønborg, Lene Holmvang.

**Project administration:** Jacob Lønborg, Lene Holmvang.

**Resources:** Hedvig Bille Andersson, Thomas Engstrøm, Peter Clemmensen, Steffen Helqvist, Erik Jørgensen, Henning Kelbæk, Frants Pedersen, Kari Saunamäki, Lene Holmvang.

**Supervision:** Jacob Lønborg, Lene Holmvang.

**Validation:** Jacob Lønborg.

**Writing – original draft:** Abdellatif El-Ahmadi, Mujahed Sebastian Abassi.

**Writing – review & editing:** Abdellatif El-Ahmadi, Mujahed Sebastian Abassi, Jacob Lønborg, Lene Holmvang.

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
