## [Decision Letter · Decision Letter 0]

10 Aug 2019

PONE-D-19-18871

Contrast induced nephropathy – a frequent and serious complication after primary percutaneous coronary intervention in patients with ST-segment elevation myocardial infarction

PLOS ONE

Dear Mr El-Ahmadi,

Thank you for submitting your manuscript to PLOS ONE. After careful consideration, we feel that it has merit but does not fully meet PLOS ONE’s publication criteria as it currently stands. Therefore, we invite you to submit a revised version of the manuscript that addresses the points raised during the review process.

We would appreciate receiving your revised manuscript by Sep 24 2019 11:59PM. To enhance the reproducibility of your results, we recommend that if applicable you deposit your laboratory protocols in protocols.io, where a protocol can be assigned its own identifier (DOI) such that it can be cited independently in the future. For instructions see: http://journals.plos.org/plosone/s/submission-guidelines#loc-laboratory-protocols

We look forward to receiving your revised manuscript.

Kind regards,

Corstiaan den Uil

Academic Editor

PLOS ONE

Journal Requirements:

1. In ethics statement in the manuscript and in the online submission form, please provide additional information about the patient records used in your retrospective study. Specifically, please ensure that you have discussed whether all data were fully anonymized before you accessed them and/or whether the IRB or ethics committee waived the requirement for informed consent. If patients provided informed written consent to have data from their medical records used in research, please include this information.

3. Please include your tables as part of your main manuscript and remove the individual files. Please note that supplementary tables (should remain/ be uploaded) as separate "supporting information" files

Reviewers' comments:

Reviewer's Responses to Questions

**Comments to the Author**

1. Is the manuscript technically sound, and do the data support the conclusions?

Reviewer #1: Yes

Reviewer #2: Yes

2. Has the statistical analysis been performed appropriately and rigorously? 

Reviewer #1: Yes

Reviewer #2: Yes

3. Have the authors made all data underlying the findings in their manuscript fully available?

Reviewer #1: Yes

Reviewer #2: Yes

4. Is the manuscript presented in an intelligible fashion and written in standard English?

Reviewer #1: Yes

Reviewer #2: Yes

5. Review Comments to the Author

Reviewer #1: I have read with interest the manuscript regarding CIN STEMI patients with undergoing primary PCI. The topic of CIN in STEMI patients undergoing primary PCI is of growing interest, as this entity is associated with adverse outcomes, and the findings may bear some clinical implications.

While interesting, several issues need to be further clarified and discussed :

1. The authors state in the introduction "…… Previous studies on the incidence, risk factors and prognostic importance of CIN in patients with STEMI treated with primary PCI have included patients from randomized studies with various in and exclusion criteria. Thus, these studies do not represent the full spectrum of patients with STEMI and there are no data on the importance of CIN in an all-comer STEMI population……" This is not accurate at all. The main finding in the manuscript is that CIN was associated with adverse short and long term outcomes. The findings are well known and have been extensively described in previous cohorts, including large retrospectives registries. For examples similar findings were described by a large registry of over 2000 STEMI patients by Margolis et al ( Journal of nephrology, 2018,31:423-428). Please discuss, and compare to the findings in that study.

2. The definition of CIN used ( 25% increase in serum creatinine within 72 hours of admission ) is somehow outdated. Recent data recommends the utilization of consensus criteria to define acute kidney injury , most updated is the KDIGO criteria. Please discuss this , and compare the limitations of CIN definition to KDIGO definition ( as mentioned by the cohort of Margolis et al, see above….) .

3. Quite surprisingly , the was no difference in contrast volume between patients with vs. without CIN. How can the authors explain this ? As such contrast volume itself was not independently associated with CIN…..

4. With regard to the above- Recent data suggest that contrast volume/eGFR ratio is a more accurate marker of CIN . Please calculate, add to table 1 (and regression model if significant ) and discuss.

5. It is known today that AKI in STEMI patients is in fact multifactorial and related to hemodynamic , inflammatory parameters in addition to the effect of contrast alone ( Shacham et al. Canadian j cardiology, 2015;31:1240-1244). Indeed patients with CIN had higher rate of cardiogenic shock, longer time from symptom onset and lower LVEF. Please cite and discuss.

6. Admission glucose levels were also demonstrated to be associated with CIN following primary PCI ( Shacham et al, cardiorenal medicine 2015;5:191-198) what were the admission glucose levels within the two groups?

Please cite and discuss

Reviewer #2: I have read carefully the manuscript entitled " Contrast induced nephropathy – a frequent and serious complication after primary percutaneous coronary intervention in patients with ST-segment elevation myocardial infarction" by Abdellatif El-Ahmadi and his colleagues. This is a well written manuscript that deals with an important topic of the prevalence and clinical significance of renal function deterioration of patients with STEMI undergoing primary PCI. The relative lack in data on this subject makes this manuscript's topic important and clinically relevant. Nevertheless I have serious concerns regarding its methodology and findings.

Major comment:

The increase in serum creatinine in patients with STEMI should be defined as AKI-acute kidney injury, rather than CIN. In this population, AKI may be related to many causes other than CIN including hemodynamic instability, cholesterol emboli, medications etc. indeed , presentation with cardiogenic shock and CPR were more frequent in patients with AKI.

Is there any data regarding fluid administration following PCI?

Is there any data regarding renal function after 72 hours? In the majority of patients with CIN renal function tends to recover and this may change the outcome

What was the percentage of patients who required renal replacement therapy?

There is no data regarding medical therapy at discharge-it may be suggested that patients with reduced renal function may be less frequently treated with guideline based medical therapy

6. PLOS authors have the option to publish the peer review history of their article (what does this mean?). If published, this will include your full peer review and any attached files.

Reviewer #1: No

Reviewer #2: No

---

## [Author Response · Author response to Decision Letter 0]

4 Nov 2019

First of all we would like to thank you for your time and your wonderfull comments on our paper. We have included the answers directly after your questions. I hope you are satisfied with our answers.

Reviewer #1: 

I have read with interest the manuscript regarding CIN STEMI patients with undergoing primary PCI. The topic of CIN in STEMI patients undergoing primary PCI is of growing interest, as this entity is associated with adverse outcomes, and the findings may bear some clinical implications.

While interesting, several issues need to be further clarified and discussed:

1. The authors state in the introduction "…… Previous studies on the incidence, risk factors and prognostic importance of CIN in patients with STEMI treated with primary PCI have included patients from randomized studies with various in and exclusion criteria. Thus, these studies do not represent the full spectrum of patients with STEMI and there are no data on the importance of CIN in an all-comer STEMI population……" This is not accurate at all. The main finding in the manuscript is that CIN was associated with adverse short and long term outcomes. The findings are well known and have been extensively described in previous cohorts, including large retrospectives registries. For examples similar findings were described by a large registry of over 2000 STEMI patients by Margolis et al ( Journal of nephrology, 2018,31:423-428). Please discuss, and compare to the findings in that study.

Answer: The fact that Margolis study have chosen 48h as the inclusion time may explain the low incidence of AKI. Compared to both the present study and many earlier studies such as Tsai (ref 3), Marcos (ref 4) and Narula (ref 8). Our inclusion time was 72h since the creatinine levels are known to rise up to 3 days after procedure.

2. The definition of CIN used ( 25% increase in serum creatinine within 72 hours of admission ) is somehow outdated. Recent data recommends the utilization of consensus criteria to define acute kidney injury , most updated is the KDIGO criteria. Please discuss this , and compare the limitations of CIN definition to KDIGO definition ( as mentioned by the cohort of Margolis et al, see above….).

Answer: We would like to state that others also have used 25% increase as in the article by Matthew S. Davenport Contrast Material–induced Nephrotoxicity and Intravenous Low-Osmolality Iodinated Contrast Material.

It is not exactly defined that the increase in serum creatinine must be 25% or any higher percentages and we are aware of others using higher percentages. In the KDIGO criteria there are 3 stages of AKI, stage 1 using 1,5-1,9 increase from baseline creatinine and stage 3 using more 3 times increase from baseline creatinine, and we translate this as a severity score, therefore 25% increase could presumably be the first point in developing the AKI and gradually getting worse with increasing creatinine levels. The authors themselves state that the score should be considered as a severity scoring.

Furthermore the “Consensus Guidelines for the Prevention of Contrast Induced Nephropathy” from Canadian Association of Radiologist stated that “The commonest definitions in use are an increase in serum creatinine (SCr) of >25% of baseline value”. We therefore chose this value as it was the same initial value that were commonest in Denmark.

Whether to use 25% increase or higher does not change the fact that our two groups shows differences ie higher long term mortality. We have included a comment on this in our discussion.

3. Quite surprisingly , the was no difference in contrast volume between patients with vs. without CIN. How can the authors explain this ? As such contrast volume itself was not independently associated with CIN…..

Answer: That contrast volume, preexisting diabetes and procedure length were not independently associated with AKI in the present population may be explained by several factors. Earlier studies reported much higher average use of contrast media volume among patients developing AKI (153-378 ml) compared to the 111-114 ml used in the present study (see ref 6,8 and 17).

4. With regard to the above- Recent data suggest that contrast volume/eGFR ratio is a more accurate marker of CIN . Please calculate, add to table 1 (and regression model if significant ) and discuss.

Answer: We have calculated the contrast volume/eGFR ratio and as you stated we have also found it significant in our univariate analysis. Therefore we need to include it in our regression model, therefore doing the whole statistical analysis from start, to make sure that all other values are exact. We need aprox. 14 days more for this. This will be uploaded as a change in both the tables and new manuscript.

5. It is known today that AKI in STEMI patients is in fact multifactorial and related to hemodynamic , inflammatory parameters in addition to the effect of contrast alone ( Shacham et al. Canadian j cardiology, 2015;31:1240-1244). Indeed patients with CIN had higher rate of cardiogenic shock, longer time from symptom onset and lower LVEF. Please cite and discuss.

Answer: Cardiogenic shock at admission and cardiac arrest before the procedure were not independently associated with AKI in the present study, possibly explained by the low frequency of both cardiac arrest and cardiogenic shock, and the fact that the most critically ill patients die before additional blood work-up can be performed. Data was also missing on LVEF why this had to be excluded from the multivariate analysis.

We have sited your reference.

6. Admission glucose levels were also demonstrated to be associated with CIN following primary PCI ( Shacham et al, cardiorenal medicine 2015;5:191-198) what were the admission glucose levels within the two groups?

Please cite and discuss

Answer: Unfortunately we do not have baseline glucose levels and are not able to comment on this.

We have sited your reference.

Reviewer #2: 

I have read carefully the manuscript entitled " Contrast induced nephropathy – a frequent and serious complication after primary percutaneous coronary intervention in patients with ST-segment elevation myocardial infarction" by Abdellatif El-Ahmadi and his colleagues. This is a well written manuscript that deals with an important topic of the prevalence and clinical significance of renal function deterioration of patients with STEMI undergoing primary PCI. The relative lack in data on this subject makes this manuscript's topic important and clinically relevant. Nevertheless I have serious concerns regarding its methodology and findings.

Major comment:

The increase in serum creatinine in patients with STEMI should be defined as AKI-acute kidney injury, rather than CIN. In this population, AKI may be related to many causes other than CIN including hemodynamic instability, cholesterol emboli, medications etc. indeed , presentation with cardiogenic shock and CPR were more frequent in patients with AKI.

Answer: You are indeed right about this matter. We just had the initial idea about the contrast medium being the cause and therefore the title remained. We have now changed the title to be more appropriate and everywhere in the manuscript changed CIN with AKI.

Is there any data regarding fluid administration following PCI?

Answer: Unfortunately we do not have data on fluid administration.

Is there any data regarding renal function after 72 hours? In the majority of patients with CIN renal function tends to recover and this may change the outcome.

Answer: We had the same idea about the renal function recovery, but most of the patients did not have follow-up creatinine levels so that we could elaborate on this.

What was the percentage of patients who required renal replacement therapy?

There is no data regarding medical therapy at discharge-it may be suggested that patients with reduced renal function may be less frequently treated with guideline based medical therapy

Answer: Unfortunately we do not have data on how many patients who required renal replacement therapy.

---

## [Editor Report · Decision Letter 1]

8 Nov 2019

PONE-D-19-18871R1

Acute Kidney Injury – a frequent and serious complication after primary percutaneous coronary intervention in patients with ST-segment elevation myocardial infarction

PLOS ONE

Dear Mr El-Ahmadi,

Thank you for submitting your manuscript to PLOS ONE. After careful consideration, we feel that it has merit but does not fully meet PLOS ONE’s publication criteria as it currently stands. Therefore, we invite you to submit a revised version of the manuscript that addresses the points raised during the review process.

The authors stated the following.

"We have calculated the contrast volume/eGFR ratio and as you stated we have also

found it significant in our univariate analysis. Therefore we need to include it in our regression

model, therefore doing the whole statistical analysis from start, to make sure that all other values

are exact. We need aprox. 14 days more for this. This will be uploaded as a change in both the

tables and new manuscript."

Please upload a final version of this manuscript.

We would appreciate receiving your revised manuscript by Dec 23 2019 11:59PM. To enhance the reproducibility of your results, we recommend that if applicable you deposit your laboratory protocols in protocols.io, where a protocol can be assigned its own identifier (DOI) such that it can be cited independently in the future. For instructions see: http://journals.plos.org/plosone/s/submission-guidelines#loc-laboratory-protocols

We look forward to receiving your revised manuscript.

Kind regards,

Corstiaan den Uil

Academic Editor

PLOS ONE

---

## [Author Response · Author response to Decision Letter 1]

29 Nov 2019

Please see Response to Reviewers letter at the end of the PDF for review

---

## [Editor Report · Decision Letter 2]

4 Dec 2019

Acute Kidney Injury – a frequent and serious complication after primary percutaneous coronary intervention in patients with ST-segment elevation myocardial infarction

PONE-D-19-18871R2

Dear Dr. El-Ahmadi,

We are pleased to inform you that your manuscript has been judged scientifically suitable for publication and will be formally accepted for publication once it complies with all outstanding technical requirements.

With kind regards,

Corstiaan den Uil

Academic Editor

PLOS ONE
---

## [Editor Report · Acceptance letter]

11 Dec 2019

PONE-D-19-18871R2 

Acute Kidney Injury – a frequent and serious complication after primary percutaneous coronary intervention in patients with ST-segment elevation myocardial infarction 

Dear Dr. El-Ahmadi:

I am pleased to inform you that your manuscript has been deemed suitable for publication in PLOS ONE. Congratulations! Your manuscript is now with our production department. 

With kind regards,

on behalf of

Dr. Corstiaan den Uil 

Academic Editor

PLOS ONE